# Spinal Astrocyte-Neuron Lactate Shuttle Contributes to the Pituitary Adenylate Cyclase-Activating Polypeptide/PAC1 Receptor-Induced Nociceptive Behaviors in Mice

**DOI:** 10.3390/biom12121859

**Published:** 2022-12-12

**Authors:** Yuki Kambe, Masafumi Youkai, Kohei Hashiguchi, Yoshimune Sameshima, Ichiro Takasaki, Atsuro Miyata, Takashi Kurihara

**Affiliations:** 1Department of Pharmacology, Graduate School of Medical and Dental Sciences, Kagoshima University, Kagoshima 890-8544, Japan; 2Department of Pharmacology, Graduate School of Science and Engineering, University of Toyama, Toyama 930-8555, Japan

**Keywords:** pituitary adenylate cyclase-activating polypeptide, astrocyte-neuron lactate shuttle, glycogen, L-lactate, monocarboxylate transporter, nociceptive behaviors, protein kinase C

## Abstract

We have previously shown that spinal pituitary adenylate cyclase-activating polypeptide (PACAP)/PACAP type 1 (PAC1) receptor signaling triggered long-lasting nociceptive behaviors through astroglial activation in mice. Since astrocyte-neuron lactate shuttle (ANLS) could be essential for long-term synaptic facilitation, we aimed to elucidate a possible involvement of spinal ANLS in the development of the PACAP/PAC1 receptor-induced nociceptive behaviors. A single intrathecal administration of PACAP induced short-term spontaneous aversive behaviors, followed by long-lasting mechanical allodynia in mice. These nociceptive behaviors were inhibited by 1,4-dideoxy-1,4-imino-d-arabinitol (DAB), an inhibitor of glycogenolysis, and this inhibition was reversed by simultaneous L-lactate application. In the cultured spinal astrocytes, the PACAP-evoked glycogenolysis and L-lactate secretion were inhibited by DAB. In addition, a protein kinase C (PKC) inhibitor attenuated the PACAP-induced nociceptive behaviors as well as the PACAP-evoked glycogenolysis and L-lactate secretion. Finally, an inhibitor for the monocarboxylate transporters blocked the L-lactate secretion from the spinal astrocytes and inhibited the PACAP- and spinal nerve ligation-induced nociceptive behaviors. These results suggested that spinal PAC1 receptor-PKC-ANLS signaling contributed to the PACAP-induced nociceptive behaviors. This signaling system could be involved in the peripheral nerve injury-induced pain-like behaviors.

## 1. Introduction

Transfer of L-lactate from astrocytes to neurons is activated when synaptic activity is increased, and this mechanism is now known as astrocyte–neuron lactate shuttle (ANLS), that could account for the coupling between synaptic activity and energy delivery [1]. Since L-lactate from astrocytes originates from glucose and glycogen [2], inhibition of glycogenolysis or L-lactate transport in the rat hippocampus significantly attenuates the working memory and long-term fear memory [3,4,5]. In addition, a brain-specific glycogen synthase knockout mouse showed a significant deficiency in the acquisition with an associative learning task and concomitantly, exhibited reduced activity-dependent changes in hippocampal synaptic strength [6]. It has also been reported that ANLS in amygdala is critical for the reconsolidation of cocaine memory in rats [7,8]. At the cellular level, L-lactate has been reported to potentiate N-methyl-D-aspartate (NMDA) receptor-mediated currents in cultured neurons and neurons in the nucleus of the solitary tract [9,10]. These findings suggested that ANLS could contribute to a variety of forms of neuronal plasticity in the central nervous system (CNS) [11], including plastic changes in the spinal nociceptive transmission. Indeed, mechanical allodynia evoked by pharmacogenetic activation of human M_3_ muscarinic DREADDs (designer receptors exclusively activated by designer drugs) coupled to G_q_ on astrocytes is alleviated by the inhibition of the monocarboxylate transporters (MCTs) [12]. However, it is still unclear what kinds of endogenous neurotransmitters evoke the ANLS activation in the spinal nociceptive transmission.

PACAP was originally isolated from ovine hypothalamic extracts based on its ability to stimulate adenylate cyclase in rat anterior pituitary cell cultures [13,14]. In a normal state, the PACAP-specific receptor, PAC1 receptor, which can couple with both G_s_ and G_q_ proteins, leading to the activation of the protein kinase A (PKA) and protein kinase C (PKC) [15], is particularly abundant in the CNS including the spinal dorsal horn [16,17,18,19,20], where PACAP-immunoreactive fibers are also considerably localized [21,22,23,24], and PACAP mRNA/immunoreactivity in rat dorsal root ganglia has been repeatedly shown to be markedly upregulated by peripheral nerve injury or inflammation [25,26,27,28]. These observations coupled with other lines of evidence propose that the PACAP/PAC1 receptor system could play an important role in the modulation of spinal nociceptive transmission.

We have previously demonstrated in mice that a single intrathecal injection of PACAP or a PAC1 receptor specific agonist, maxadilan (Max) [29], induced spontaneous aversive behaviors, such as licking, biting, and scratching directed toward the caudal part of the body for more than 30 min [30,31], and the aversive behaviors were followed by an induction of the mechanical allodynia, which lasted for more than 84 days [20]. However, vasoactive intestinal polypeptide (VIP), which shares VIP/PACAP (VPAC) 1 or VPAC2 receptors with PACAP, failed to induce these nociceptive behaviors [20,31]. Co-treatment of a PAC1 receptor antagonist, max.d.4 with PACAP, almost completely inhibited the induction of both aversive behaviors and the mechanical allodynia [20,31]. These results suggested a critical role of the PAC1 receptor in the nociceptive transmission. Immunohistochemical and immunoblotting studies revealed that spinal application of PACAP or Max induced upregulation of an astrocyte marker, glial fibrillary acidic protein (GFAP), at least for 84 days, and L-α-aminoadipate, an astroglial toxin, attenuated the induction of both aversive behaviors and mechanical allodynia. Interestingly, L-α-aminoadipate was still effective to reverse the mechanical allodynia even if it was intrathecal-administered 84 days after spinal PAC1 receptor stimulation [20]. These results suggest that long-lasting spinal astocytic activation underlies the PACAP/PAC1 receptor-induced aversive behaviors and mechanical allodynia. However, how spinal astrocytes contribute to these nociceptive behaviors is still unknown.

L-lactate is one of the substrates of MCTs, and MCTs are found in the brain where three isoforms—MCT1, MCT2 and MCT4—have been described. Although the distribution patterns of these MCTs are not well-known in the spinal cord, each of these isoforms is suggested to exhibit a distinct regional and cellular distribution in rodent brain. At the cellular level, MCT1 is known to be expressed by endothelial cells of microvessels, by ependymocytes as well as by astrocytes. MCT4 expression appears to be specific for astrocytes. By contrast, the predominant neuronal MCT is suggested to be MCT2 [32]. It is reported that AR-C155858 potently inhibits MCT1 and MCT2, but not MCT4 [33].

Glycogen is the single largest energy reserve in the CNS and is predominantly localized in astrocytes of the intermediate lobe of the pituitary gland [34]. Among the neurotransmitters and bioactive substances examined up until today, PACAP was suggested to have potent glycogenolytic activity with an EC_50_ value of 0.08 nM [35]. To the best of our knowledge, PACAP had the highest efficacy to induce glycogenolysis in cultured astrocytes derived from the cerebral cortex. However, it is not known whether PACAP is involved in the spinal ANLS. Therefore, we aimed to examine the possible involvement of the PACAP/PAC1 receptor system in the spinal ANLS, and also to test whether the spinal ANLS contributes to the PACAP/PAC1 receptor-induced nociceptive behaviors.

## 2. Materials and Methods

### 2.1. Animals

Male ddY mice (6~12 weeks old) or pregnant female ddY mice were purchased from Kyudo Co. Ltd. (Kumamoto, Japan) and housed under a controlled temperature (24 ± 1 °C) and humidity (55 ± 10%) with a 12-h light/dark cycle with food and water freely available.

### 2.2. Drugs

PACAP (38 amino acid form) and VIP were purchased from Peptide Institute Inc. (Osaka, Japan). Max was kindly donated by Dr. M Tajima (Shiseido, Tokyo, Japan). 1, 4-dideoxy-1, 4-imino-d-arabinitol (DAB), Bisindolylmaleimide I (GF109203X) and phorbol 12-myristate 13-acetate (PMA) were obtained from Wako Pure Chemical Industries, Ltd. (Osaka, Japan). Forskolin and AR-C155858 were from Merck Millipore (Darmstadt, Germany). H89 was from Seikagaku Co. (Tokyo, Japan). L-lactate and 8-Bromoadenosine-3′,5′-cyclic monophosphorothioate, Rp-isomer (Rp-8-Br-cAMPS) were from Sigma-Aldrich Co. LLC (St. Louis, MO, USA). 9-(tetrahydrofuran-2-yl)-9h-purin-6-amine (Sq22.536) was from Enzo Biochem Inc. (Farmingdale, NY, USA). These drugs were made up as concentrated stock solution in MilliQ water, 0.1 M acetic acid or dimethyl sulfoxide, aliquoted, and stored at −30 °C and diluted just before use. The aliquot of drugs was diluted to the desired concentration in artificial cerebrospinal fluid (ACSF: NaCl 138 mM, KCl 3 mM, CaCl_2_ 1.25 mM, MgCl_2_ 1 mM, D-glucose 1 mM) immediately prior to use. The doses for intrathecal injection of PACAP (100 pmol) and Max (50 pmol) we chose in this study were determined according to our previous reports [20,30,31].

### 2.3. Intrathecal Injection and Behavioral Observation

Intrathecal injection of Max, PACAP, DAB, L-Lactate, GF109203X and AR-C155858 was given in a volume of 5 μL by percutaneous puncture through an intervertebral space at the level of the fifth or sixth lumbar vertebra, according to a previously reported procedure [20,30,31]. An investigator, who was unaware of the drug treatment, performed all of the behavioral experiments.

Aversive behavior was evaluated according to our previous report [31]. Before intrathecal injection, mice were placed and habituated in a glass cylinder (φ14 × 18 cm) with a filter paper at the bottom for 20 min. Immediately after intrathecal injection, the mice were placed again in the same glass cylinder and the number of aversive behaviors consisting of licking, biting and scratching directed toward the caudal part of the body was counted every 1 min. The cumulative number of events was pooled for 5-min bins of observation and analyzed.

The assessment of mechanical thresholds was also carried out according to the previously described methods [20]. Briefly, mechanical sensitivity was evaluated with calibrated von Frey hairs (Stoelting, Wood Dale, IL, USA) by measuring the tactile stimulus producing a 50% likelihood of hind paw withdrawal response (50%-gram threshold), which was determined using the up–down paradigm [36].

### 2.4. Cultured Spinal Cord Astrocytes

Astrocytes were cultured as previously reported [35]. Spinal cords from postnatal day 1 and 2 mice were dispersed by 0.25% trypsin and subsequently plated on culture flasks which were previously coated with 0.75 μg/mL poly-l-lysine (Sigma-Aldrich, St. Louis, MO, USA). Cells were cultured in Dulbecco’s modified Eagle medium (Nacalai Tesque, Osaka, Japan) supplemented with 10% fetal bovine serum (Hyclone, South Logan, UT, USA) until cell density reaching to confluent. After reaching to confluent, culture flasks were shaken at 240 rpm for 6 h to remove microglia and oligodendrocyte precursor cells. Remaining astrocytes on culture flasks were then detached and subsequently plated on the appropriate culture dishes at the density of 1 × 10^4^ cells/cm^2^ and used when the cell number was reaching to the confluent again, which usually took 3 days. All the cultured astrocytes used in the current experiments were passaged 3 or 4 times. All the cell-based experiments were carried out at 37 °C.

### 2.5. Immunocytochemistry

Spinal cord astrocytes were plated on cover slips (Matsunami Glass, Tokyo, Japan). According to the reported protocols for immunocytochemistry [37], cells were fixed for 10 min with 4% paraformaldehyde, washed with phosphate-buffered saline (PBS), and incubated in PBS containing 5% bovine serum albumin and 0.3% Triton X-100 for 60 min. Cells were then incubated with antibodies specific for GFAP (Merck Millipore, AB5804, rabbit IgG, 1/1000) [38], microtubule-associated protein-2 (MAP2, Sigma-Aldrich Co. LLC, M4403, AB_477193, mouse IgG, 1/1000) [39], ionized calcium binding adaptor molecule-1 (Iba1, Wako Pure Chemical Industries, Ltd., 019-19741, AB_839504, rabbit IgG, 1/1000) [40] and 2′,3′-cyclic-nucleotide 3′-phosphodiesterase (CNPase, Abcam, ab6319, AB_2082593, mouse IgG, 1/1000) [41], followed by anti-mouse IgG and anti-rabbit IgG secondary antibodies coupled to CF488 and CF555 (Biotium, Hayward, CA, USA), respectively. Samples were mounted with CC/mount (Diagnostic BioSystems, Pleasanton, CA, USA). The fluorescent signals were observed using a fluorescence microscope (BZ-X700: Keyence, Osaka, Japan).

### 2.6. Glycogen Assay

Cultured spinal astrocytes were exposed to PACAP, VIP or Max for 1 h, and harvested by scratching. Immediately after harvest, samples were then boiled for 10 min, sonicated and centrifuged. Glycogen amount in the supernatant was measured by the Glycogen Assay Kit (BioVision Inc., San Francisco, CA, USA) according to the manufacturer’s instruction. The exposure of DAB, H89, Rp-8-Br-cAMPS, Sq22.536 or GF109203X was started 30 min before PACAP exposure. The amount of glycogen was normalized by the amount of the protein contained in the samples, which was measured by the Bradford’s assay kit (Bio-Rad, Hercules, CA, USA) according to the manufacturer’s instruction.

### 2.7. L-Lactate Assay

The culture medium was replaced Krebs–Ringer buffer (NaCl 135 mM, KCl 5 mM, CaCl_2_ 1 mM, MgSO_4_ 1 mM, KH_2_PO_4_ 0.4 mM, D-glucose 5.5 mM, HEPES 20 mM). The astrocytes were then exposed to PACAP, PMA or forskolin for 60 min, and conditioned buffer was harvested (0~60 min, designated as “early phase”). Subsequently, fresh Krebs–Ringer buffer with the drugs was added on the cultured astrocytes again and harvested the conditioned buffer for another 60 min (60~120 min, designated as “late phase”). The time course was determined according to the previous paper which suggested DAB inhibited L-lactate secretion in 60~180 min after the treatment, but not in 0~60 min [42]. L-Lactate amount in the conditioned Krebs–Ringer buffer was measured by the L-lactate Assay Kit (BioVision Inc.) according to the manufacturer’s instruction. The exposure of DAB or GF109203X was started 30 min before PACAP treatment.

### 2.8. Spinal Nerve Ligation (SNL) Surgery

To produce peripheral neuropathic pain, SNL was carried out as described [43,44]. Briefly, a midline incision was made in the skin of the back at the L4-S2 levels and the right paraspinal muscles were separated from the spinous processes, facet joints and transverse processes at the L4–S1 levels. The right L4–L5 transverse processes were removed, and the right L4 and L5 spinal nerves were ligated tightly with 8-0 silk thread.

### 2.9. Ethics

All experiments in the present study were approved by the Experimental Animal Research Committee of Kagoshima University (Approval number: MD18003). All animal experiments were performed following the ARRIVE guidelines and the National Institutes of Health guide for the care and use of laboratory animals. The work described in the present study was carried out following The Code of Ethics of the World Medical Association for animal experiments (http://ec.europa.eu/environment/chemicals/lab_animals/legislation_en.htm, accessed on 5 October 2022) and Uniform Requirements for manuscripts submitted to Biomedical Journals (http://www.icmje.org, accessed on 5 October 2022).

### 2.10. Statistical Analysis

Experimental data are expressed as mean ± SEM. For mechanical threshold analyses, we employed the Mann–Whitney U-test for single comparisons or the Friedman test followed by the Steel or Steel–Dwass test for multiple comparisons. For other analyses, single comparisons were made using the Student’s two-tailed unpaired *t*-test, and for multiple comparisons, one-way analysis of variance followed by the Dunnett or Tukey test was used. *p* < 0.05 was considered statistically significant.

## 3. Results

### 3.1. PACAP/PAC1 Receptor-Induced Nociceptive Behaviors Were Attenuated by the Inhibition of Glycogen Phosphorylase with DAB

We first examined a possible involvement of spinal ANLS in the PACAP/PAC1 receptor-induced nociceptive behaviors in mice. After a single intrathecal administration of Max (50 pmol) or PACAP (100 pmol), aversive behaviors such as licking and biting gradually appeared within 0~5 min, reached a plateau around 15~20 min, and was maintained for at least 30 min. The co-injection of DAB, a glycogen phosphorylase (PYGB) inhibitor, dose-dependently (1~100 pmol) attenuated the development of the Max- or PACAP-induced aversive behaviors (Figure 1a,c). In addition, a single intrathecal administration of Max or PACAP markedly decreased mechanical threshold (induction of mechanical allodynia) from day 1 (after the cessation of the aversive behaviors), and this decrease persisted at least for 21 days after the administration. The co-injection of DAB with Max or PACAP also dose-dependently attenuated the induction of mechanical allodynia by Max or PACAP (Figure 1b,d).

### 3.2. Suppression of the PAC1 Receptor-Evoked Nociceptive Behaviors by DAB Was Reversed by Intrathecal Co-Injection of L-Lactate

We then asked whether the inhibition of the PAC1 receptor-induced nociceptive behaviors by DAB could be reversed by the co-administration of exogenous L-lactate. Intrathecal injection of L-lactate (1 nmol) in combination with Max (50 pmol) + DAB (10 pmol) significantly reversed the anti-aversive and anti-allodynic effects of DAB on the PAC1 receptor-induced nociceptive behaviors (Figure 2a,b). Similarly, intrathecal supplementation of L-lactate (1 nmol) also blocked the inhibitory effects of DAB (100 pmol) on the PACAP-induced aversive behaviors and mechanical allodynia (Figure 2c,d). 

### 3.3. Possible Involvement of PKC in the PACAP/PAC1 Receptor-Evoked Glycogenolysis in Cultured Spinal Cord Astrocytes

We prepared cultured astrocytes from newborn mice spinal cords and confirmed that most cells were suggested to be astrocytes, <1% cells were oligodendrocytes, and no neurons or microglial cells were detected, when visualized with immunocytochemistry for specific markers of individual cells (Appendix A). In order to identify the receptor subtypes involved in the PACAP-evoked glycogenolysis in the spinal astrocytes, we exposed PACAP, VIP or Max and determined glycogen amount in the cells. Exposure of PACAP (0.0001~10 nM) or Max (0.1~10 nM) dose-dependently decreased glycogen amount (Figure 3a,b), but VIP failed to decrease the glycogen amount even at the concentration of 10 nM (Figure 3c). This PACAP-induced glycogenolysis was completely abolished by the DAB (1 mM) (Figure 3d). Next, we co-applied GF109203X as a PKC inhibitor, H89 or Rp-8-Br-cAMPS as a PKA inhibitor, or Sq22.536 as an adenylate cyclase (AC) inhibitor with PACAP. We found that GF109203X (5 μM) significantly inhibited the PACAP-evoked glycogenolysis (Figure 3e), but neither PKA inhibitors (Figure 3f) nor the AC inhibitor (Appendix A) affected the glycogenolysis. 

### 3.4. PKC Is Crucial for the PACAP/PAC1 Receptor-Evoked L-Lactate Secretion in the Cultured Spinal Astrocytes

To investigate whether L-lactate secretion from astrocytes is increased by the PACAP/PAC1 receptor/PKC pathway, we measured the L-lactate amounts in the supernatants from the cultured spinal astrocytes after drug treatment. PACAP (1 nM) significantly increased L-lactate amounts in the supernatant in both 0~60 min (early phase) and 60~120 min (late phase) after the exposure (Figure 4a,b). PMA (100 nM), a PKC activator, significantly elevated L-lactate secretion in the late phase (Figure 4c,d), while forskolin (5 μM), an AC activator, failed to enhance L-lactate secretion (Figure 4e,f). In addition, GF109203X (5 μM) significantly inhibited the PACAP-activated L-lactate secretion in both phases (Figure 4g,h). Furthermore, the inhibition of glycogenolysis by DAB significantly decreased the L-lactate amount in the supernatant from the astrocytes particularly in the late phase (Figure 4i,j).

### 3.5. Blockade of the PACAP/PAC1 Receptor-Induced Nociceptive Behaviors by the PKC Inhibitor, GF109203X

In order to investigate whether PACAP/PAC1 receptor-induced nociceptive behaviors were also mediated through the PKC pathway, the effects of GF109203X were investigated. Both the aversive behaviors and the mechanical allodynia evoked by PACAP were dose-dependently inhibited by the simultaneous injection of GF109203X at a concentration range from 1 to 100 pmol (Figure 5a,b).

### 3.6. Pharmacological Inhibition of Monocarboxylate Transporters Attenuated the PACAP/PAC1 Receptor-Induced Nociceptive Behaviors

We further asked whether L-lactate transport participated in the PACAP/PAC1 receptor-induced nociceptive behaviors. In our cultured spinal astrocytes, AR-C155858 (1 μM) significantly inhibited L-lactate secretion evoked by PACAP at both phases (Figure 6a,b). Simultaneous intrathecal injection of AR-C155858 (1 nmol) with PACAP significantly inhibited the development of the aversive behaviors (Figure 6c) and mechanical allodynia (Figure 6d). Intriguingly, intrathecal administration of AR-C155858 transiently alleviated the mechanical allodynia at 7 days after intrathecal injection of PACAP (Figure 6e).

### 3.7. Reversal of Spinal Nerve Injury (SNL)-Induced Mechanical Allodynia by the PYGB Inhibitor, DAB and MCT Inhibitor, AR-C155858

Several lines of evidence indicate that the spinal PACAP/PAC1 receptor could play an important role in the induction of peripheral nerve injury-induced neuropathic pain. For instance, the development of neuropathic pain-like behaviors induced by SNL was shown to be abrogated in PACAP-deficient mice [28], and SNL-induced mechanical allodynia was ameliorated by the treatment with intrathecal PACAP_6-38_, a peptide antagonist of PAC1/VPAC2 [45]. Recently, we developed several small-molecule PAC1 receptor selective antagonists [46,47,48] and showed that both intrathecal and systemic (peroral) administration of these small-molecule antagonists significantly reversed SNL-induced mechanical allodynia [47,48]. Thus, we finally tested the effects of DAB and AR-C155858 on SNL-induced mechanical allodynia. In this study, behavioral studies were performed with mice at 14 days after SNL (Figure 7). The SNL induced marked mechanical hypersensitivity of the hindpaw ipsilateral to the ligation and intrathecal administration of DAB (100 pmol) significantly reversed the mechanical allodynia (Figure 7a). Furthermore, intrathecal treatment of AR-C155858 (1~3 nmol) produced dose-dependent decreases in the mechanical allodynia (Figure 7b).

## 4. Discussion

In this study, we have further characterized the nociceptive behaviors induced by the spinal PAC1 receptor activation and made the following findings: (1) PACAP could be an endogenous inducer for spinal ANLS activation; (2) PAC1/PKC signaling played an important role in the PACAP-evoked spinal ANLS; (3) the spinal ANLS activation contributed to PACAP-evoked nociceptive behaviors such as aversive behaviors and mechanical allodynia; and (4) the spinal ANLS activity would be involved in the generation of SNL-induced mechanical allodynia.

In the present study, we showed that the simultaneous intrathecal injection of DAB, the inhibitor of glycogenolysis, blocked the development of aversive behaviors and mechanical allodynia evoked by Max or PACAP, and this inhibition by DAB was reversed by the combinational injection of L-lactate. These results suggested that the ANLS activation may have a significant role in the nociceptive behaviors by PACAP/PAC1 receptor activation. Chronic pain may need maladaptive synaptic plasticity in neural circuits [49,50]. An in vivo electrophysiological study showed that DAB inhibited the long-term potentiation by tetanus stimulus in hippocampal Schaffer collateral-area CA1, and this impairment by DAB was rescued with the exogenous L-lactate [4]. Furthermore, L-lactate alone was shown to potentiate α-3-hydroxy-5-methyl-4-isoxazole propionic acid (AMPA) receptor- or NMDA receptor-mediated inward currents, increase intracellular calcium, and also upregulate the markers for neuronal activation [4,9,10]. Thus, PACAP/PAC1 receptor-evoked spinal ANLS might potentiate AMPA/NMDA receptors-mediated responses and induce long-term potentiation, which might lead to aversive behaviors and long-lasting mechanical allodynia. Regarding the initiation and maintenance of inflammatory and neuropathic pain, recent progresses also focus on the critical roles of astrocytes in the spinal cord and brain [51,52]. Previously, we showed that intrathecal injection of Max or PACAP induced rapid and prolonged upregulation of the spinal GFAP expression level in parallel with the aversive behaviors and long-lasting mechanical allodynia, and simultaneous application of L-α-aminoadipate, an astrocyte toxin, with Max or PACAP markedly suppressed the aversive behaviors and the mechanical allodynia [20,31]. Interestingly, it is suggested that glycogen metabolism is associated with maturation of astrocytes, in which the expressional levels of GFAP and glycogen synthase or phosphorylase are increased in parallel [53]. In accordance with this notion, our present study revealed that ANLS activation indeed contributed to both development and maintenance of PACAP/PAC1 receptor-evoked nociceptive behaviors, since simultaneous intrathecal injection of DAB and AR-C155858 could prevent the development of the PACAP-induced aversive behaviors and mechanical allodynia. Furthermore, intrathecal administration of DAB and AR-C155858 at 7 or 14 days after intrathecal PACAP injection or SNL surgery, respectively, ameliorates the evoked-mechanical allodynia, which suggested that spinal ANLS activation might contribute to the pathogenic mechanism of peripheral nerve injury-induced neuropathic pain. Although further study is needed to clarify the persistency of ANLS activation and its role in the long-lasting mechanical allodynia, these results suggested the therapeutic potential of the ANLS inhibitors for chronic pain treatments.

Glycogen contained in cultured spinal cord astrocytes was significantly decreased by the Max or PACAP, but not VIP, exposure. These results suggested that the PAC1 receptor, but not the VPAC1 or VPAC2 receptor, was responsible for activating glycogenolysis. In rat atrial myocytes, a PKC inhibitor peptide reduced the PACAP-induced K_ATP_ currents without affecting the VIP-induced K_ATP_ currents [54], indicating selective coupling of PKC with PAC1 receptor. Indeed, in our present study, PACAP-evoked glycogenolysis was significantly inhibited by the PKC inhibitor, GF109203X. These results suggested that the PAC1 receptor/PKC signaling pathway is important for the glycogenolysis activation. Previous reports which observed glycogenolysis in chick brain or cultured mouse cortical astrocytes suggested that the cAMP/PKA signaling pathway phosphorylated glycogen phosphorylase and evoked glycogenolysis [55,56]; but in our cultured spinal astrocytes, both PKA inhibitors and the AC inhibitor failed to prevent the PACAP-evoked glycogenolysis. Although the reason why PACAP-evoked glycogenolysis is not sensitive to the AC and PKA inhibitors is currently unknown, this may be due to the absence of the coupling of the PACAP/PAC1 receptor to G_s_ protein in the spinal astrocytes. Further study is required to verify this possibility. 

We found that developments of the PACAP/PAC1 receptor-induced aversive behaviors and mechanical allodynia were significantly attenuated by the PKC inhibition with GF109203X. In cultured astrocytes, PACAP and PMA significantly enhanced L-lactate secretion. In addition, GF109203X and DAB significantly inhibited the PACAP-enhanced L-lactate secretion. These results proposed that PKC and a downstream glycogenolysis were important in the PACAP-enhanced L-lactate secretion and nociceptive behaviors. Many lines of evidence strongly suggested the significance of PKC in inflammatory and neuropathic pain [57]. Double labeling immunofluorescence showed that phospho-PKC-δ co-localized with astrocytic markers in the spinal cord after spinal nerve injury [58]. These studies suggest that astrocytic PKC contributes to the nociceptive behaviors. Although there is little evidence about the contribution of PKC in astrocytic MCT activation, it has been previously reported that the PKC activation resulted in an increase in the levels of MCT1 and MCT4 and the release of L-lactate in in vitro skeletal muscle cells (RD cells), while the PKA activation resulted in a significant decrease in MCT1 expression in the RD cells as well as endothelial cells [59,60,61]. Importantly, the mechanical allodynia by selective astrocytic G_q_-PKC activation was ameliorated by pre-treatment with a MCT inhibitor, 4-CIN (α-cyano-4-hydroxycinnamate) [12]. These results indicate that PKC-induced L-lactate secretion through MCTs might be involved in the PACAP/PAC1 receptor-evoked nociceptive behaviors.

It may be worth noting our previous data here that intrathecal injection of the PKA inhibitor, Rp-8-Br-cAMPS, could ameliorate the PACAP-induced aversive behaviors [31]. Currently, it may be difficult to precisely reconcile with the present observation that the PKA inhibitor failed to prevent PACAP-evoked glycogenolysis in the spinal astrocytes, but, for example, we could speculate that neuronal PKA signaling would also contribute to the expression of the nociceptive behaviors. However, further extensive studies are necessary to uncover the interaction of PKC and PKA signaling in the PACAP/PAC1 receptor-induced nociceptive behaviors.

It may be noteworthy that DAB significantly inhibited the L-lactate secretion in the late phase after PACAP exposure, while the inhibitory effect was relatively weak in the early phase. Our results may be consistent with a previous report that DAB significantly attenuated the L-lactate secretion for 60~180 min after the hypoglycemia on the astrocytes/neurons mixed culture, while an equivalent amount of L-lactate was secreted irrespective of the presence of DAB for 0~60 min; even DAB completely inhibited glycogenolysis at both timings [42]. Thus, these observations may indicate that the secreted L-lactate in the late phase originated from glycogenolysis, and the secretion in the early phase may be partly derived from the readily releasable pool of L-lactate in the cultured spinal astrocytes. However, the delayed in vitro inhibition of L-lactate secretion by DAB is difficult to reconcile with the relatively rapid effect on the PACAP-induced aversive behaviors, and further studies under the experimental setting close to in vivo condition would be necessary in the future to answer the discrepancy.

## 5. Conclusions

The present study suggested that the interaction between spinal dorsal horn neurons and astrocytes evoked by PACAP/PAC1 receptor-induced ANLS activation could be critically involved in the development and maintenance of the nociceptive behaviors. The signaling pathway linking PAC1 receptor activation and ANLS might be at least partially mediated by the PKC pathway (Figure 8). Although verifying the possible involvement of the PACAP/PAC1 receptor signaling in human pain requires further rigorous studies, targeting spinal ANLS may provide a new opportunity to treat intractable chronic pain.

PACAP administered intrathecally to the spinal dorsal horn, or endogenously released PACAP, activates PAC1 receptors on astrocytes. The activated PAC1 receptors stimulate glycogenolysis via PKC signaling and facilitate L-lactate secretion through MCTs. The released L-lactate acts on dorsal horn neurons and contributes to prolonged pain behaviors.

## Figures and Tables

**Figure 1 biomolecules-12-01859-f001:**
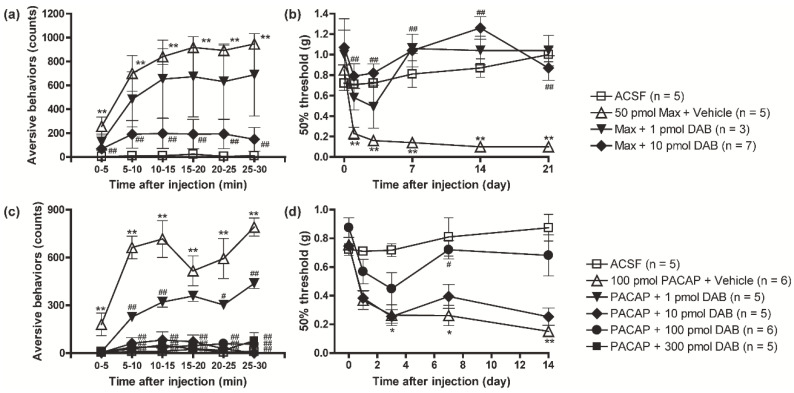
A critical role of glycogenolysis in PACAP/PAC1 receptor-evoked nociceptive behaviors. Simultaneous intrathecal injection of DAB (1 or 10 pmol), a glycogenolysis inhibitor, with Max (50 pmol), blocked the induction of aversive behaviors (**a**) and mechanical allodynia (**b**). Simultaneous intrathecal injection of DAB (1~300 pmol) with PACAP (100 pmol) prevented the induction of aversive behaviors (**c**) and mechanical allodynia (**d**). * *p* < 0.05 and ** *p* < 0.01 when compared with ACSF data. ^#^
*p* < 0.05 and ^##^
*p* < 0.01 when compared with Max + Vehicle in (**a**,**b**) or PACAP + Vehicle in (**c**,**d**). Statistical significance was evaluated by the Tukey test for (**a**,**c**), and Steel test for (**b**,**d**).

**Figure 2 biomolecules-12-01859-f002:**
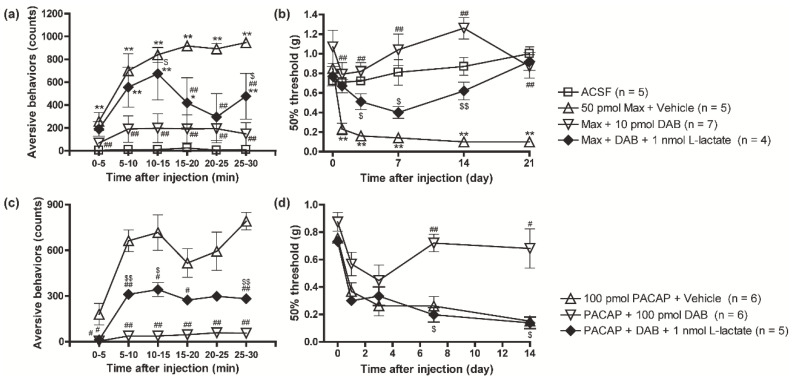
Simultaneous injection of L-lactate reversed the inhibitory effects of DAB on the PAC1 receptor-induced nociceptive behaviors. Intrathecal co-administration of L-lactate (1 nmol) with DAB (10 or 100 pmol) reinstated Max (**a**,**b**) and PACAP (**c**,**d**) induced aversive behaviors (**a**,**c**) and mechanical allodynia (**b**,**d**). * *p* < 0.05 and ** *p* < 0.01 when compared with ACSF data. ^#^
*p* < 0.05 and ^##^
*p* < 0.01 when compared with Max + Vehicle in (**a**,**b**) or PACAP + Vehicle in (**c**,**d**). ^$^
*p* < 0.05 and ^$$^
*p* < 0.01 when compared with DAB-treated data. Statistical significance was evaluated by the Tukey test for (**a**,**c**), and Steel test for (**b**,**d**).

**Figure 3 biomolecules-12-01859-f003:**
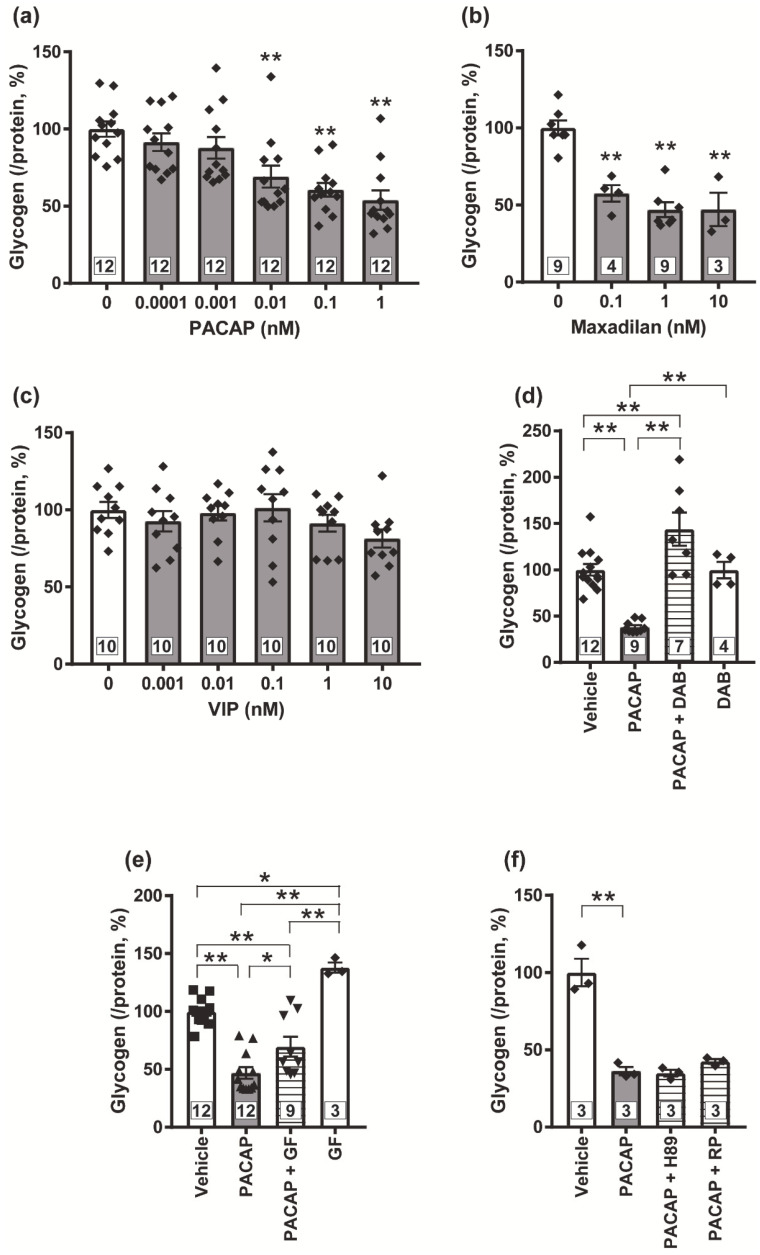
PACAP/PAC1 receptor/PKC signaling pathway induced glycogenolysis in cultured spinal cord astrocytes. Cultured astrocytes were exposed to PACAP (0.0001~1 nM) (**a**), Max (0.1~10 nM) (**b**) or VIP (0.001~10 nM) (**c**), and glycogen amounts contained in the astrocytes were measured 60 min after the exposure. Cultured astrocytes were pre-incubated with DAB (1 mM, (**d**)), GF109203X (GF, 5 μM, (**e**)), H89 (10 μM, (**f**)), or Rp-8Br-cAMPS (RP, 100 μM, (**f**)) for 30 min prior to PACAP (1 nM) exposure, and glycogen amounts contained in the astrocytes were measured 60 min after the PACAP exposure. * *p* < 0.05 and ** *p* < 0.01 when compared with 0 nM (Vehicle) data in (**a**–**c**), and compared groups are indicated above in (**d**–**f**). Statistical significance was evaluated by the Dunnett test for (**a**–**c**), and Tukey test for (**d**–**f**). Exact sample sizes are indicated in the graphs.

**Figure 4 biomolecules-12-01859-f004:**
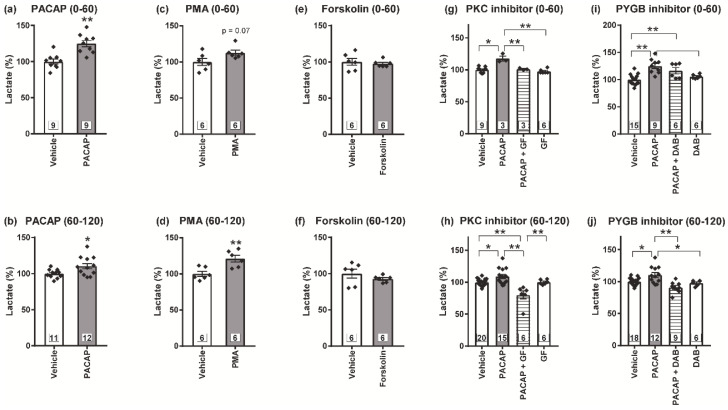
A crucial role of PACAP/PAC1 receptor/PKC-induced glycogenolysis in the L-lactate secretion from cultured spinal astrocytes. After the treatment with PACAP (1 nM) (**a**,**b**), PMA (100 nM) (**c**,**d**) or forskolin (5 μM) (**e**,**f**), the supernatants from the cultured spinal astrocytes were separately harvested during 0–60 min (**a**,**c**,**e**) or 60–120 min (**b**,**d**,**f**), and L-lactate amounts were measured. Cultured astrocytes were pre-incubated with GF109203X (GF, 5 μM, (**g**,**h**)) or DAB (1 mM, (**i**,**j**)) for 30 min prior to PACAP exposure, and then exposed to PACAP (1 nM) for 120 min. Culture supernatants were separately harvested during 0–60 min (**g**,**i**) or 60–120 min (**h**,**j**) after the PACAP addition, and L-lactate amounts were measured. * *p* < 0.05 and ** *p* < 0.01 when compared with vehicle data in (**a**–**f**), and compared groups are indicated above in (**g**–**j**). Statistical significance was evaluated by the Student’s *t*-test for (**a**–**f**), and Tukey test for (**g**–**j**). Exact sample sizes are indicated in the graphs.

**Figure 5 biomolecules-12-01859-f005:**
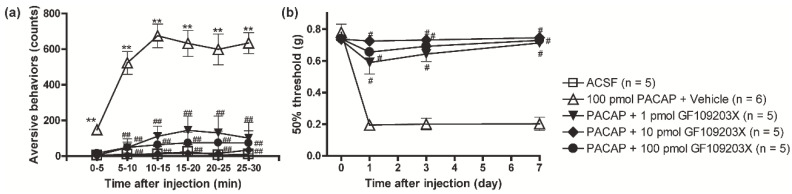
The importance of PKC in the PACAP/PAC1 receptor-induced nociceptive behaviors. Simultaneous intrathecal injection of GF109203X (1~100 pmol) with PACAP (100 pmol) dose-dependently reduced the PACAP-induced aversive behaviors (**a**) and mechanical allodynia (**b**). ** *p* < 0.01 when compared with ACSF data. ^#^
*p* < 0.05 and ^##^
*p* < 0.01 when compared with PACAP + Vehicle data. Statistical significance was evaluated by the Tukey test for (**a**), and Steel test for (**b**).

**Figure 6 biomolecules-12-01859-f006:**
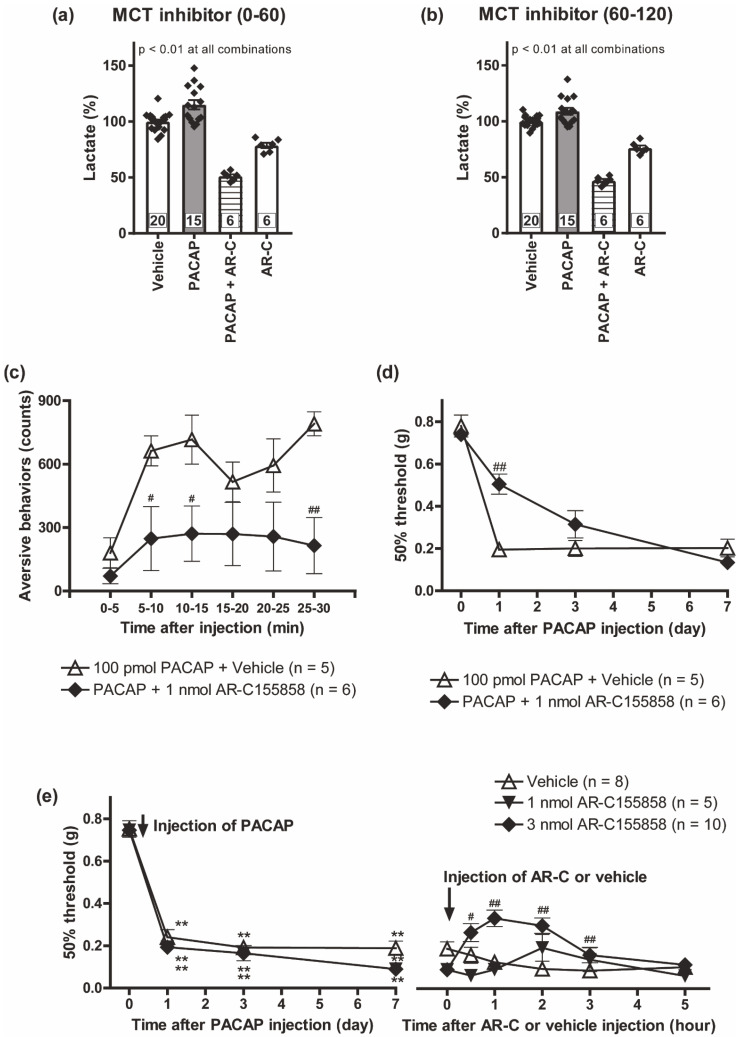
Spinal MCTs contributed to both induction and maintenance of the PACAP/PAC1 receptor-induced nociceptive behaviors. Cultured spinal astrocytes were pre-incubated with AR-C155858 (AR-C, 1 μM) 30 min prior to PACAP exposure, and then exposed to PACAP (1 nM) for 120 min. The culture supernatants were separately harvested for 0–60 min (**a**) or 60–120 min (**b**) after the PACAP addition, and L-lactate amounts in the culture supernatants were measured. ** *p* < 0.01 (compared groups are indicated above). Exact sample sizes are indicated in the graphs. Intrathecal co-administration of AR-C155858 (1 nmol) with PACAP reduced the expression of the aversive behaviors (**c**) and mechanical allodynia (**d**). ^#^
*p* < 0.05 and ^##^
*p* < 0.01 when compared with PACAP + Vehicle data. (**e**) Transient reversal of the PACAP-evoked mechanical allodynia by intrathecal AR-C155858 injection. AR-C155858 was injected 7 days after intrathecal PACAP injection. ** *p* < 0.01 when compared with pre-PACAP data. ^#^
*p* < 0.05 and ^##^
*p* < 0.01 when compared with vehicle-treated control. Statistical significance was evaluated by the Tukey test for (**a**,**b**), Student *t*-test for (**c**), Mann–Whitney U test for (**d**) and Steel test for (**e**).

**Figure 7 biomolecules-12-01859-f007:**
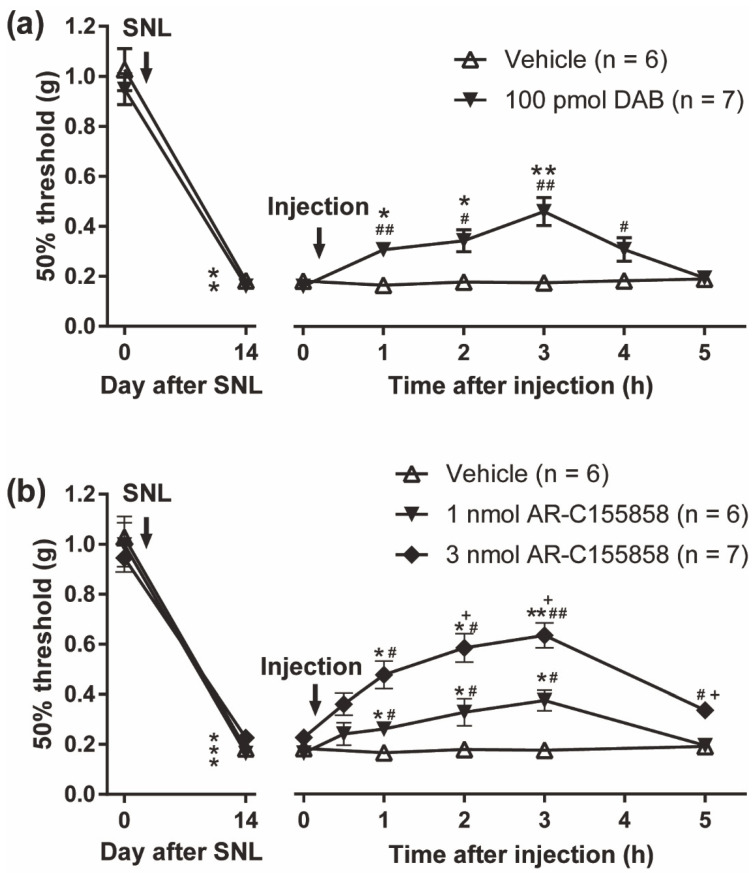
Requirement of spinal ANLS in the maintenance of SNL-induced mechanical allodynia. (**a**) Transient reversal of SNL-evoked mechanical allodynia by intrathecal DAB injection. DAB was injected 14 days after SNL surgery. * *p* < 0.05 and ** *p* < 0.01 when compared with Day 0 or time 0 data. ^#^
*p* < 0.05 and ^##^
*p* < 0.01 when compared with vehicle-treated control. (**b**) Significant anti-allodynic effects of intrathecal AR-C155858 injection on SNL-induced neuropathic pain model mice. AR-C155858 was injected 14 days after SNL surgery. * *p* < 0.05 and ** *p* < 0.01 when compared with Day 0 or time 0 data. ^#^
*p* < 0.05 and ^##^
*p* < 0.01 when compared with vehicle-treated control. ^+^
*p* < 0.05 when compared with 1 nmol AR-C155858-treated group. Statistical significance was evaluated by the Mann–Whitney U test for comparison between pre- and post-SNL surgery, and Steel–Dwass test for comparison between pre- and post-drug injection.

**Figure 8 biomolecules-12-01859-f008:**
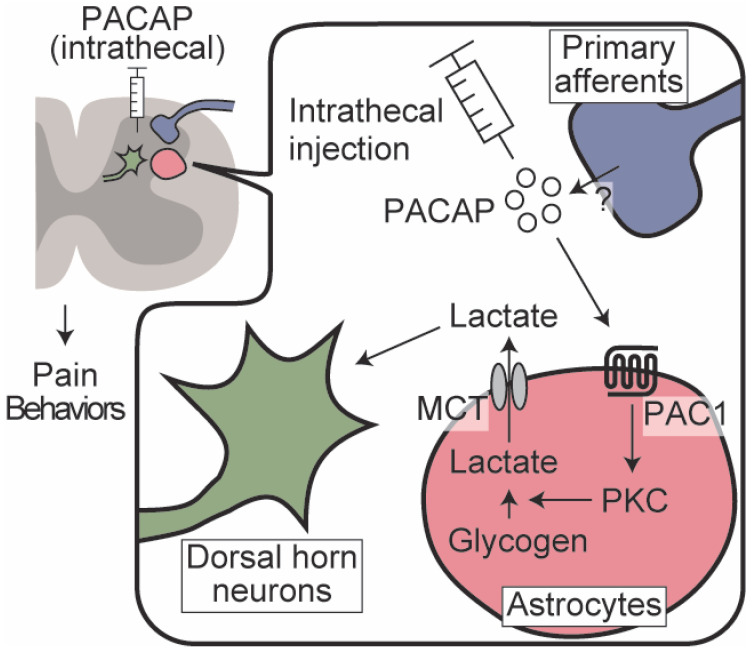
Schematic illustration of the present study.

## Data Availability

The data that support the findings of this study are available on reasonable request from the corresponding author.

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
