# Peer review of "Spinal Astrocyte-Neuron Lactate Shuttle Contributes to the Pituitary Adenylate Cyclase-Activating Polypeptide/PAC1 Receptor-Induced Nociceptive Behaviors in Mice"

_biomolecules, 2022, doi:10.3390/biom12121859_

Round 1
Reviewer 1 Report
ANSL is a modern hypothesis that astrocytes respond to increased neuronal activity by increasing the rate of glucose uptake, glycolysis and lactate release into the extracellular space. However, there is little information about the factors involved in these processes. The manuscript entitled „Spinal astrocyte-neuron lactate shuttle contributes to the pituitary adenylate cyclase-activating polypeptide/PAC1 receptor-induced nociceptive behaviors in mice” written by Yuki Kambe , Masafumi Youkai , Kohei Hashiguchi , Yoshimune Sameshima , Ichiro Takasaki , Atsuro Miyata , Takashi Kurihara makes an effort to broaden this knowledge, as well as being a continuation of earlier research conducted by the co-authors of this article.
However, below are some notes that should be included/changed in the manuscript:
- Please explain the abbreviations in the text, e.g. NMDA, GFAP, H89, Rp-8-Br-cAMPS, Sq22.536, GF109203X, AMPA, please complete. Due to the large number of abbreviations, I suggest creating a table with full names for text clarity and easy reading.
- The data regarding the aproval of the Local ethics committee should be completed.
Materials and Methods
- 2.2. I.t. injection and behavioral observation
Please expand the abbreviation „i.t.” as any abbreviation. In addition, whether i.t. not change to IT because "intrathecal" is one word (for that matter).
What substances were applied intrathecally,?
Maybe combine chapter 2.2 with 2.3. or reword for better readability? (just a suggestion)
- 2.5. Immunocytochemistry-
It is advisable to specify the species for each primary antibody (MAP2, Iba1, CNPase). What antibody combinations were used? Please complete the data.
What methods of stain specificity control were used?
The manuscript lacks photographic documentation of immunocytochemical studies. Please complete.
Results
This chapter needs redrafting. By the standards of the journal, the Results section should contain only precise data obtained in the experiment described in this paper. Therefore, descriptions of studies already published, suggestions, reflections are redundant in this chapter. Citing other studies at this chapter causes confusion. These sentences or paragraphs should be moved to the Introduction or Discussion section. For example:
Page 11, line 207 –„… These results suggested that spinal ANLS has a critical role in the PACAP/PAC1 receptor-induced nociceptive behaviors.”
Page 12, line 226 „..suggesting that PAC1 receptor is primarily involved in the glycogenolysis. Interestingly, this pharmacological characteristic of glycogenolysis was very similar to that of the PACAP-induced nociceptive behaviors in mice reported previously [19,30]. „…This PACAP-induced glycogenolysis was completely abolished by the DAB (1 mM) (Figure 3d), indicating that the glycogenolysis by PACAP would be mediated by PYGB. Since PAC1 receptor can couple with both Gs and Gq proteins, leading to the activation of the protein kinase A (PKA) and PKC, respectively [39],….”.
Page 17, line 291 - L-lactate is one of the substrates of MCTs, and MCTs are found in the brain where three isoforms MCT1, MCT2 and MCT4 — have been described. Although the distribution patterns of these MCTs are not well-known in the spinal cord, each of these isoforms is suggested to exhibit a distinct regional and cellular distribution in rodent brain. At the cellular level, MCT1 is known to be expressed by endothelial cells of microvessels, by ependymocytes as well as by astrocytes. MCT4 expression appears to be specific for astrocytes. By contrast, the predominant neuronal MCT is suggested to be MCT2 [40]. It is reported that AR- C155858 potently inhibits MCT1 and MCT2, but not MCT4 [41].
Author Response
Response: Thank you for taking your valuable time to review our paper. We are confident that the manuscript is even better after correcting the areas you pointed out. The changes are noted in red in the manuscript.
ANSL is a modern hypothesis that astrocytes respond to increased neuronal activity by increasing the rate of glucose uptake, glycolysis and lactate release into the extracellular space. However, there is little information about the factors involved in these processes. The manuscript entitled „Spinal astrocyte-neuron lactate shuttle contributes to the pituitary adenylate cyclase-activating polypeptide/PAC1 receptor-induced nociceptive behaviors in mice” written by Yuki Kambe , Masafumi Youkai , Kohei Hashiguchi , Yoshimune Sameshima , Ichiro Takasaki , Atsuro Miyata , Takashi Kurihara makes an effort to broaden this knowledge, as well as being a continuation of earlier research conducted by the co-authors of this article.
However, below are some notes that should be included/changed in the manuscript:
- Please explain the abbreviations in the text, e.g. NMDA, GFAP, H89, Rp-8-Br-cAMPS, Sq22.536, GF109203X, AMPA, please complete. Due to the large number of abbreviations, I suggest creating a table with full names for text clarity and easy reading.
Response: Thank you for your valuable comments. I have now included the full name of the words you pointed out, and also created a list of abbreviations at the beginning of the revised manuscript on lines 34-62.
- The data regarding the aproval of the Local ethics committee should be completed.
Response: Thank you for your important remarks. This study was approved in advance by the local GMO and Animal Experimental Committee, which was omitted from the description, so we have added it in the chapter 2.9 of the revised manuscript.
Materials and Methods
- 2.2. I.t. injection and behavioral observation
Please expand the abbreviation „i.t.” as any abbreviation. In addition, whether i.t. not change to IT because "intrathecal" is one word (for that matter).
What substances were applied intrathecally,?
Maybe combine chapter 2.2 with 2.3. or reword for better readability? (just a suggestion)
Response: Thank you for your constructive comments. We have swapped chapters 2.2. and 2.3. to make it easier to read. Also, in the new chapter 2.3., we have listed the drugs administered intrathecally. We have also decided not to abbreviate i.t. throughout the manuscript.
- 2.5. Immunocytochemistry-
It is advisable to specify the species for each primary antibody (MAP2, Iba1, CNPase). What antibody combinations were used? Please complete the data.
What methods of stain specificity control were used?
The manuscript lacks photographic documentation of immunocytochemical studies. Please complete.
Response: Thank you so much for your kind suggestion for our immunocytochemistry. As the reviewer suggested, we added the originated species of each primary antibody in the chapter 2.5 (Immunohistochemistry). Antibodies we used were frequently used in previous studies. Therefore, we added the references (references #38 ~ #41) to show the specificity of each antibody. The description of the immunocytochemistry has been mentioned in the chapter 3.3. of the original version of our manuscript.
Results
This chapter needs redrafting. By the standards of the journal, the Results section should contain only precise data obtained in the experiment described in this paper. Therefore, descriptions of studies already published, suggestions, reflections are redundant in this chapter. Citing other studies at this chapter causes confusion. These sentences or paragraphs should be moved to the Introduction or Discussion section. For example:
Response: Thank you for your important remarks. We have made the changes described below to the revised Result session.
Page 11, line 207 –„… These results suggested that spinal ANLS has a critical role in the PACAP/PAC1 receptor-induced nociceptive behaviors.”
Response: I deleted this sentence because a similar sentence was already available in the discussion session.
Page 12, line 226 „..suggesting that PAC1 receptor is primarily involved in the glycogenolysis. Interestingly, this pharmacological characteristic of glycogenolysis was very similar to that of the PACAP-induced nociceptive behaviors in mice reported previously [19,30]. „…This PACAP-induced glycogenolysis was completely abolished by the DAB (1 mM) (Figure 3d), indicating that the glycogenolysis by PACAP would be mediated by PYGB. Since PAC1 receptor can couple with both Gs and Gq proteins, leading to the activation of the protein kinase A (PKA) and PKC, respectively [39],….”.
Response: I deleted the former part („..suggesting that…previously [19,30] and …indicating that….by PYGB) because a similar sentence was already available in the discussion session. The latter sentence (Since…[39]…) is now available in the revise Introduction section on lines 84-85.
Page 17, line 291 - L-lactate is one of the substrates of MCTs, and MCTs are found in the brain where three isoforms MCT1, MCT2 and MCT4 — have been described. Although the distribution patterns of these MCTs are not well-known in the spinal cord, each of these isoforms is suggested to exhibit a distinct regional and cellular distribution in rodent brain. At the cellular level, MCT1 is known to be expressed by endothelial cells of microvessels, by ependymocytes as well as by astrocytes. MCT4 expression appears to be specific for astrocytes. By contrast, the predominant neuronal MCT is suggested to be MCT2 [40]. It is reported that AR- C155858 potently inhibits MCT1 and MCT2, but not MCT4 [41].
Response: This sentence is also now available in the revised Introduction section on lines 107-113.

Reviewer 2 Report
This manuscript is scientifically very sound, but may not be of interest to many readers. My only suggestion is that the authors read through the manuscript once again to be sure that the language is clear and to the point.
For example, I.t. seems to me to be an odd abbreviation for intrathecal, when iv is the accepted abbreviation for intravenous.
Another example, lines 264ff what is meant by "differentially harvested?"
Nonetheless, the conclusions seem to be well supported by the experiments.
Author Response
Response: Thank you for taking your valuable time to review our paper. We are confident that the manuscript is even better after correcting the areas you pointed out. The changes are noted in red in the manuscript.
This manuscript is scientifically very sound, but may not be of interest to many readers. My only suggestion is that the authors read through the manuscript once again to be sure that the language is clear and to the point.
Response: Thank you so much for your kind suggestion. We have read thoroughly once again and corrected several words (shown in red).
For example, I.t. seems to me to be an odd abbreviation for intrathecal, when iv is the accepted abbreviation for intravenous.
Response: Thank you for your constructive comments. We have decided not to abbreviate i.t. throughout the manuscript.
Another example, lines 264ff what is meant by "differentially harvested?"
Response: Thank you for your important remarks. We changed “differentially” to “separately” in the new Figure 4 and 6 legends.
Nonetheless, the conclusions seem to be well supported by the experiments.

Round 2
Reviewer 1 Report
The manuscript entitled „Spinal astrocyte-neuron lactate shuttle contributes to the pituitary adenylate cyclase-activating polypeptide/PAC1 receptor-induced nociceptive behaviors in mice” written by Yuki Kambe , Masafumi Youkai , Kohei Hashiguchi , Yoshimune Sameshima , Ichiro Takasaki , Atsuro Miyata , Takashi Kurihara still needs minor corrections.
Results:
The text from line: 259-260, 283-285, 304-310, 328-329, 373-374 should be placed in the appropriate place in the Discussion. Please change.
Author Response
The manuscript entitled „Spinal astrocyte-neuron lactate shuttle contributes to the pituitary adenylate cyclase-activating polypeptide/PAC1 receptor-induced nociceptive behaviors in mice” written by Yuki Kambe , Masafumi Youkai , Kohei Hashiguchi , Yoshimune Sameshima , Ichiro Takasaki , Atsuro Miyata , Takashi Kurihara still needs minor corrections.
Response: Thank you for taking your valuable time to review our paper. We are confident that the manuscript is even better after correcting the areas you pointed out. The changes are noted in red in the revised-2 manuscript.
Results:
The text from line: 259-260, 283-285, 304-310, 328-329, 373-374 should be placed in the appropriate place in the Discussion. Please change.
Response: Thank you for your very constructive comments. The lines: 259-260, 283-285, 304-310, 328-329, 373-374 in the result section are deleted, and now available at lines: 389-391, 411-412, 426-427, 433-434, 437 in the discussion section of the newly revised manuscript.